

# Effects of the 2014 Major Baltic Inflow on methane and nitrous oxide dynamics in the water column of the Central Baltic Sea

Jukka-Pekka Myllykangas[1], Tom Jilbert[1], Gunnar Jakobs[1,2], Gregor Rehder[3], Jan Werner[3], and Susanna Hietanen[1]

[1]Department of Environmental Sciences, University of Helsinki, P.O. Box 65, FIN-00014 University of Helsinki, Finland
[2]Technical University of Denmark, Frederiksborgvej 399, 4000 Roskilde, Denmark
[3]Leibniz Institute for Baltic Sea Research Warnemünde (IOW), Seestraße 15, D-18119 Rostock, Germany

*Correspondence to:* JP Myllykangas (jukka-pekka.myllykangas@helsinki.fi)

**Abstract.** In late 2014, a large, oxygen-rich salt water inflow entered the Baltic Sea and caused considerable changes in deep water oxygen concentrations. We studied the effects of the inflow on the concentration patterns of two greenhouse gases, methane and nitrous oxide, during the following year (2015) in the water column of the Gotland Basin. Methane which had previously accumulated in anoxic deep waters was largely removed from the Eastern Gotland Basin during the year, predomi-

nantly due to oxidation following turbulent mixing with the oxygen-rich inflow. In contrast nitrous oxide, which was previously absent from deep waters, accumulated in deep waters due to enhanced nitrification following the inflow. A transient extreme accumulation of nitrous oxide was observed in the deep waters of the Eastern Gotland Basin towards the end of 2015, when deep waters turned anoxic again and sedimentary denitrification was induced. The Western Gotland Basin gas biogeochemistry was not affected by the inflow.

## 1  Introduction

The Baltic Sea is a shallow, semi-enclosed brackish water body. It receives large fresh water inputs from the rivers along its coast, but also exchanges saline water with the North Sea through the narrow Danish straits, principally via the Darss Sill and the Drogden Sill (Fig. 1). This leads to a semi-permanent stratification between relatively fresh surface waters and denser, more saline deep waters. Although intermediate-depth water masses in the southern areas of the Baltic Sea are ventilated frequently,

deep waters of the central Baltic are renewed only during Major Baltic Inflow (MBI) events, during which large amounts of saline oxygen-rich water enter the Baltic through the Danish straits over a short period of time (Schinke and Matthäus, 1998). These events require a specific sequence of weather conditions, occur exclusively in winter and have been occurring approximately once a decade in the recent past (Gräwe et al., 2015).

Due to the semi-permanent stratification, waters below the halocline of the Baltic Sea are typically anoxic (Carstensen

et al., 2014) and contain large inventories of reduced compounds produced by microbial and abiotic reactions in the absence of oxygen (Neumann et al., 1997). Many biogeochemical processes are also active in the hypoxic boundary layers close to the halocline (Yakushev et al., 2007; Dellwig et al., 2010). When oxygen is introduced by MBIs, large quantities of the previously accumulated reduced compounds in the deep waters are subsequently oxidized (Reissmann et al., 2009) and new



redox fronts develop between new and old water masses (Schmale et al., 2016). As such, MBIs have a strong influence on many biogeochemical processes in the Baltic Sea.

In this study we focus on two gases that are strongly influenced by the spatial distribution of hypoxia and anoxia in the Baltic Sea: methane ($CH_4$) (Schmale et al., 2012) and nitrous oxide ($N_2O$) (Hietanen et al., 2012). Both are important greenhouse

gases, which also have effects on atmospheric chemistry (Crutzen, 1974; Cicerone and Oremland, 1988). Ambient oxygen concentrations regulate the microbial processes involved in the production and consumption of $CH_4$ and $N_2O$. $CH_4$ is produced in sediments in vast quantities by a unique group of archaea called methanogens (Balch et al., 1979). Methanogenesis is the lowest energy-yield pathway of the anaerobic decay of organic matter, which typically occurs when all other electron acceptors have been depleted (Thauer, 1998). The primary methanogenesis pathway in marine sediments is $CO_2$ reduction

by hydrogenotrophic methanogens, while fermentative acetotrophic methanogenesis is the dominant pathway in freshwater sediments (Whiticar et al., 1986). Typically, most of the produced $CH_4$ is consumed in anaerobic and aerobic processes in both sediments and the water column before it can escape to the atmosphere (Reeburgh, 2007; Knittel and Boetius, 2009). $N_2O$ is primarily produced as a side product in nitrification and as an intermediate product in denitrification (Anderson and Levine, 1986). The main biological pathways of $N_2O$ production are highly dependent on the oxygen conditions and the availability of

organic matter, nitrite ($NO_2^-$) and nitrate ($NO_3^-$) (Ward, 2013). In seas, nitrification has been considered the primary pathway of $N_2O$ production (Freing et al., 2012), but recent studies have suggested that the role of incomplete denitrification in the oceanic oxygen minimum zones might have been previously underestimated (Babbin et al., 2015; Ji et al., 2015). $N_2O$ is consumed exclusively under anoxic conditions during denitrification (Goreau et al., 1980; Wrage et al., 2001).

In the Baltic Sea, surface water concentrations of both gases typically exceed equilibrium with the sea-level atmosphere,

indicating supersaturation and an efflux of $CH_4$ and $N_2O$ from surface waters (Bange, 2006; Gülzow et al., 2013). However, the sub-halocline profiles of the two gases differ markedly. Deep waters are typically strongly enriched in $CH_4$ below the halocline during stagnation periods (Bange et al., 2009; Jakobs et al., 2013), while $N_2O$ is usually absent (Brettar and Rheinheimer, 1991). Increased anthropogenic nutrient loading during the last century has been linked to the enhanced production of both gases in the Baltic Sea (Bange, 2006). In the global $CH_4$ budget, the flux from the oceans to the atmosphere is estimated at

approximately 2 % of total $CH_4$ emissions (Judd et al., 2002; Reeburgh, 2007), but estimates as high as 10 % have also been presented (Grunwald et al., 2009). Shallow coastal areas and estuaries are an important component, potentially contributing up to 75 % of the total oceanic $CH_4$ flux (Bange et al., 1994). In the case of $N_2O$, the oceans are a much more important source in the global atmospheric budget, contributing up to 25 % of emissions (Nevison et al., 2003). However, there are large uncertainties in the role of coastal zones and estuaries, ranging from 11 % to 60 % of the total $N_2O$ flux from the marine

environment (Bange et al., 1996; Seitzinger and Kroeze, 1998).

In late December of 2014 a large MBI occurred, during which 198 $km^3$ of water, containing 4 Gt of salt and $2.04 \times 10^6$ t of oxygen, entered the Baltic through the Darss sill (Fig. 1). The inflow was the third largest MBI observed since 1880 (Gräwe et al., 2015; Mohrholz et al., 2015) and caused large changes in the dissolved oxygen concentrations throughout the southern and central Baltic Sea (Mohrholz et al., 2015). The inflow strongly impacted the vertical distribution of $CH_4$ in the Gotland Basin (Schmale et al., 2016), which is the largest sub-basin of the Baltic Sea. Both advective processes (displacement and



dilution of old $CH_4$-rich deep waters by the $CH_4$-poor inflow water) and aerobic oxidation of $CH_4$ (stimulated by mixing at the contact between these water masses) contributed to a decline in $[CH_4]$ during 2015.

Here we present a broader investigation of the spatial and temporal evolution of both $CH_4$ and $N_2O$ concentrations following the inflow, along a transect of sites in both the Eastern and Western Gotland Basin. We discuss the roles of advection and microbial processing in the observed distributions, and the timescales of change in biogeochemical processes in response to the perturbation caused by the MBI.

## 2 Materials and Methods

Samples were collected on six cruises that took place between March and December 2015 on R/V Aranda and R/V Salme. Sampling covered the whole Gotland Basin, with three stations in the Eastern Gotland Basin (EGB): BY10, BY15 and BY20, and two stations in the western Basin (WGB): BY32 and BY38 (Fig. 1). The coordinates of all stations are listed in Table S1.

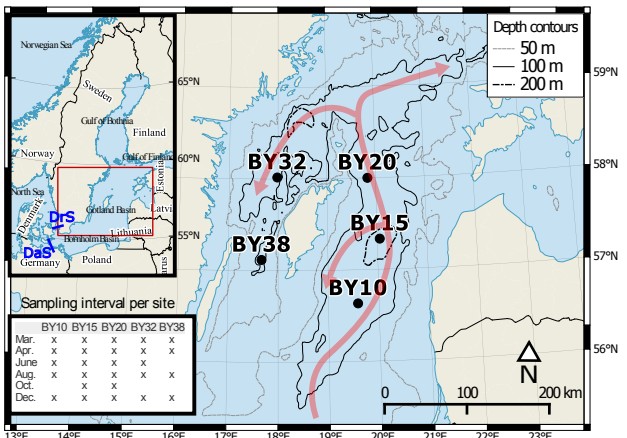

**Figure 1.** Map of the study area showing the five sampling locations in the central Baltic Sea. The red arrows show the prevailing sub-halocline circulation of the central Baltic Sea, redrawn from Meier (2007). The sub-halocline circulation approximates the expected route of Major Baltic Inflows (MBIs) along the transect of study locations. The inset shows the connection of the Baltic Sea to the North Sea via the Danish Straits (DaS = Darss Sill, DrS = Drogden Sill). The box in the lower left corner shows the sampling times for each site during the year 2015.

Water samples were collected with rosette water samplers with 12 Niskin bottles and salinity, temperature and oxygen data were generated with attached CTD probes (bottle sizes 1.7 and 5 L, CTD probes SeaBird SBE 19+ and SeaBird SBE 911+, for R/V Salme and R/V Aranda, respectively). Supplementary nutrient, oxygen and hydrogen sulphide ($H_2S$) monitoring data were provided by the Finnish Environment Institute (SYKE), Swedish Meteorological and Hydrological Institute (SMHI) and Tallinn University of Technology (TTU).

Dissolved gas samples were collected in triplicate by filling 60 mL plastic syringes directly from the Niskin bottles. The water volume in the syringe was reduced to 30 mL, a 3-way stopcock was attached to the syringe, and 31 mL of 5.0 purity $N_2$




gas was injected via the stopcock to create a headspace. Syringes were left at 20°C for 30 minutes and then vigorously shaken for three minutes. At least 25 mL of the headspace was transferred into a dry syringe and then injected into a pre-evacuated 12 mL gas-tight glass vial with a 4 mm butyl rubber stopper (LabCo Exetainer™ model 839W).

The gas samples were analysed within five months from sampling using an Agilent Technologies 7890B gas chromatograph with a flame ionization detector (FID) for $CH_4$ and an electron capture detector (ECD) for $N_2O$, a 2.4 m Hayesep Q column with 1/8" connection, 80/100 mesh range and a 1.0 mL sample loop. Carrier gases were nitrogen and helium at 21 mL min$^{-1}$ flow rates, for $N_2O$ and $CH_4$ respectively. The oven temperature was 60°C, and detection temperatures for FID and ECD were 250°C and 300°C respectively. The samples were injected using a Gilson GX-271 auto sampler. Raw peak area data were

converted to mole fraction (ppm) using linear calibration with standards. For $CH_4$, a three-point calibration was used (0.46, 5 and 47 ppm), consisting of a standard gas mixture (AGA Gas AB, Lidingö, Sweden) with 5 ppm ± 2 % $CH_4$, which was diluted 1:11 with 5.0 purity $N_2$ to create the low standard. For the high standard, a 1000 ppm ± 2 % $CH_4$ gas mixture (Air Products PLC, Surrey, United Kingdom) was diluted 1:21 with 5.0 $N_2$. For $N_2O$, a two-point calibration (0.10 and 1.1 ppm) was used, consisting of a standard gas mixture (AGA Gas AB, Lidingö, Sweden) with 1.1 ppm ± 5 % $N_2O$, which was then diluted 1:11

with 5.0 $N_2$. Standard series were analysed prior to each analysis sequence (length of sequence was 40–120 samples) and fitted linearly for each sequence separately to correct for between-series drift. Standards containing 5 ppm $CH_4$ and 1.1 ppm $N_2O$ were analysed after every 10 samples to monitor within-series drift (observed to be negligible).

Total *in situ* dissolved gas concentrations ($C_{tot}$) in mol/L were calculated from measured wet mole fraction (ppm) values in the headspace gas considering Henry's Law as per Buller (2008). In Equation 1, the first term on the right side represents

the contribution to $C_{tot}$ of dissolved gas evolved into the headspace during equilibration, while the second term represents the contribution from gas remaining in the dissolved phase:

$$C_{tot} = \frac{(X_{HS} \times P_{atm} \times V_{HS})}{(R \times T \times V_{aq})} + \beta \times X_{HS} \times P_{atm} \tag{1}$$

where $X_{HS}$ is the mole fraction of the gas in the headspace in ppm, $P_{atm}$ the pressure in the headspace in atm (1), $V_{HS}$ and $V_{Aq}$ the syringe headspace and water volume in mL respectively, $R$ is the gas constant (0.08206 L atm K$^{-1}$ mol$^{-1}$), $T$ the

temperature in Kelvin after equilibration (293 K), and $\beta$ is the Bunsen solubility coefficient for each sample at *in situ* salinity, 1 atm pressure and 293 K, as detailed in Wiesenburg and Guinasso (1979) for $CH_4$ and Weiss and Price (1980) for $N_2O$.

Pre-evacuated Exetainers contain small amounts of air, which contaminates the gas samples, standards and blanks (Sturm et al., 2015). Because the concentration of $CH_4$ and $N_2O$ in air is variable, the contamination also introduced imprecision to sample data. We calculated cut-off concentrations for both gases, below which measured values were considered indistin-

guishable from those of blank pre-evacuated Exetainers. The determination of the cut-off value took into account both the mean contamination of blank samples and various sources of imprecision in sample data (Fig S1). The cut-off concentrations are approximately equivalent to 9–19 nM and 4–6 nM for $CH_4$ and $N_2O$, respectively (exact values are salinity-dependent). Relative standard deviation (RSD) of all reported data above the cut-off is <5 %.





## 3 Results

All stations exhibited strong salinity stratification on all sampling occasions, with clear differences between the surface and bottom water salinities (Fig. 2a). The halocline was typically at 60–80 m depth. We observed a net increase in the bottom water

salinities at BY15, BY20, BY32 and BY38 from March to December. Absolute bottom water salinities decreased along the expected route of the inflow (Fig. 1; Fig. 2a).

As expected, the earliest major impact of the inflow on deep water oxygen was observed at the southernmost station BY10, where oxygen was detected already in March (up to 84 μM in the bottom water, Fig. 2b). At this site, the oxygenated zone in the bottom water expanded upwards until June, with concentrations between 90 and 120 μM. However, [$O_2$] started to decrease

considerably in August and remained low thereafter, dropping to 5 μM in December. No $H_2S$ was detected at any time at BY10. [$CH_4$] remained low throughout, with the highest value of 97 nM measured in April at 125 m depth (Fig. 2c). [$N_2O$] was between 6 and 22 nM, with the lowest concentration found in June at 100 m depth, and the highest in December in the deepest 144 m sample (Fig. 2d).

At the deepest station, BY15, in the central EGB, oxygen was also detected in the bottom water in March, although the

concentration was lower than at BY10 (50 μM, Fig. 2b). In the early part of the year, BY15 had a completely anoxic midwater layer from 100–175 m (Fig. 2b), which contained up to 21 μM $H_2S$ in April. The anoxic layer diminished over time, and had completely disappeared by August, as the oxic zone in the deep waters expanded vertically over this period. The highest bottom water [$O_2$] of 177 μM was measured in April (Fig. 2b). [$CH_4$] was relatively low throughout, except for the anoxic layer, where concentrations of up to 217 nM were measured in March. By December, bottom water [$O_2$] had decreased to below 10 μM and

[$CH_4$] of up to 158 nM were measured in the bottom water. Also, slightly elevated [$CH_4$] and [$N_2O$] were detected in October and December at 90–125 m (Fig. 2c). Minimal amounts of $N_2O$ were found within the anoxic midwater layer of BY15, but concentrations of 18–20 nM could be detected around its upper and lower boundaries (Fig. 2d). In October however, very high [$N_2O$] (877 nM) was measured at 225 m. The extreme concentrations had decreased by December, but still remained relatively high (41 nM at 225 m and 236 m) compared to previous months (Fig. 2d).

At the northernmost station of the EGB, BY20, [$O_2$] were very low or zero below 70–80 m (Fig. 2b) and $H_2S$ was found in the bottom water on all sampling occasions, with concentrations up to 33 μM in August. Bottom water was devoid of $N_2O$ and [$CH_4$] remained high (299–525 nM) (Fig. 2c), except in October when bottom water [$CH_4$] decreased to 91 nM and [$N_2O$] increased to 151 nM (Fig. 2d). Concentrations of both gases had returned to typical values by December. In August, a transient midwater [$N_2O$] maximum of 36 nM was found at 100 m depth and in October and December [$CH_4$] of up to 230 nM were

observed at 90 m depth.

In the WGB at BY32 and BY38, no oxygen was detected below the halocline at any time (Fig. 2b), and $H_2S$ was present in all bottom water samples. Bottom water [$CH_4$] displayed large variation between 308 and 726 nM (Fig. 2c), and [$N_2O$] was consistently below the cut-off value, indicating values close to zero (Fig. 2d). At both stations, a strong shoaling of the halocline could be observed over the course of the year (Fig. 2a), with the $CH_4$-enriched deep water mass expanding from 125 to 80 m at BY32, and from 90 to 70 m at BY38.





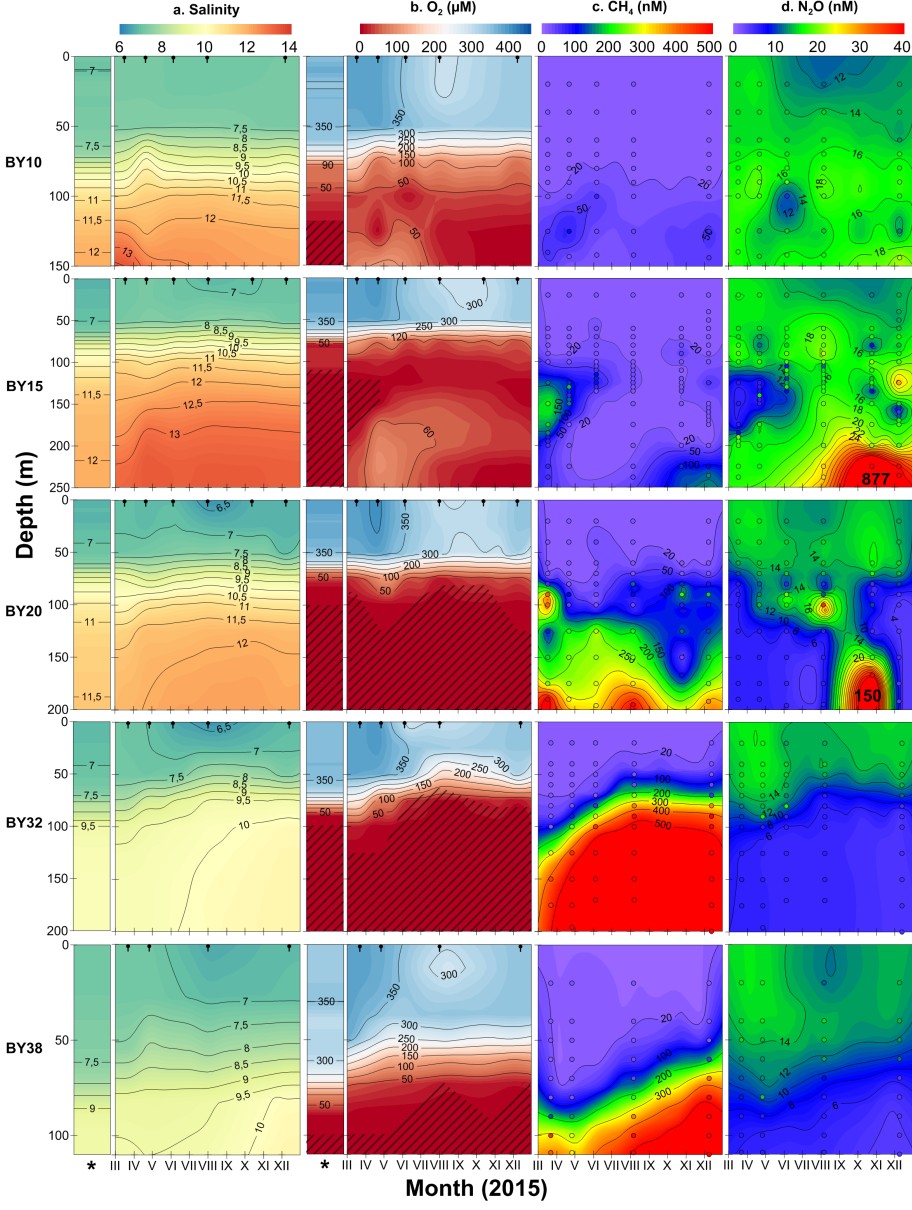

**Figure 2.** Bilinearly interpolated profiles of a) CTD salinity, b) CTD oxygen (µM), c) [CH$_4$] (nM) and d) [N$_2$O] (nM) from all sampling stations at all sampling times of the survey. The x-axis covers the period 17 March 2015 to 13 December 2015. Precise sampling dates in the CTD profiles are indicated by the black notches at the top of each panel. The narrower bars marked with an asterisk represent the salinity and oxygen conditions in June 2014, prior to the MBI. Circles in columns c) and d) represent gas sampling depths on the corresponding dates of sampling. The color of the circle infill indicates measured concentrations according to the same scaling as the interpolation. Note that all CH$_4$ and N$_2$O data below the cut-off values of peak area during GC analysis (approximately equivalent to 9–19 nM and 4–6 nM for [CH$_4$] and [N$_2$O], respectively. See Fig. S1 for details) were set to the cut-off value prior to interpolation. The presence of H$_2$S is denoted with the black shading in column b). Note also that the deepest gas sampling depth varied slightly between sampling times and that in June samples below 135 m were lost at BY15.



## 4 Discussion

### 4.1 Spatial impacts of the MBI

Major Baltic Inflows usually progress northwards through the EGB, encircle the island of Gotland counterclockwise, and finally move southwards through the WGB (Meier, 2007; Lessin et al., 2014). However, the MBI of 2014 had not reached the WGB by the end of 2015. Although bottom water salinities increased in the WGB during 2015 (Fig. 2a), the increase was continuous throughout the year and not accompanied by changes in oxygen conditions. The changes observed in the salinity of the WGB early in the year were therefore likely unrelated to the inflow. Based on 16 years of monitoring data (Fig S2), the WGB experiences annual to multi-annual cycles of shoaling and deepening of the halocline, which are related to downward erosion of the halocline by winter storms (Reissmann et al., 2009). The resulting changes in salinity and oxygen conditions in the 60–100 m depth interval dictate the distribution of $CH_4$ (Jakobs et al., 2014) and $N_2O$ in the WGB. $[CH_4]$ is higher below the halocline, while $[N_2O]$ is higher above it (Fig. 2c; Fig. 2d).

In the EGB, in contrast, large impacts of the 2014 MBI could be observed throughout 2015. Already by March, oxic water and noticeable changes in salinity were detected in the deepest part of the Eastern Gotland Basin (site BY15). These initial signals were a combination of new inflowing water and water pushed out of the Bornholm Basin ahead of the inflow (Schmale et al., 2016), as also occurred during the MBI of 2003 (Feistel et al., 2003). Over the following months, a distinct mass of saline, oxic inflow water accumulated in the EGB. At BY10 and BY15, oxygen was present below the halocline for much of 2015. However, $[O_2]$ declined again towards the end of the year (Fig. 2b). Such a decline was also observed following the 2003 MBI (Walter et al., 2006), and indicates that the capacity of MBIs to ventilate the Baltic is short-lived. Introduced oxygen is expected to be consumed simultaneously by a range of electron donors, including hydrogen sulphide ($H_2S$), ammonium ($NH_4^+$), reduced forms of manganese and iron, and $CH_4$, which are all present in the stagnant deep waters. Both the physical effects of the inflow (displacement of water masses) and the subsequent evolution of redox conditions throughout 2015, had strong impacts on the distribution of $CH_4$ and $N_2O$ at the EGB sites (BY10, BY15 and BY20).

### 4.2 $CH_4$ dynamics in the EGB

Upon arrival of an MBI into the EGB, the former bottom water mass is typically displaced vertically upwards and northwards (Reissmann et al., 2009). This displacement of $CH_4$-rich stagnant deep water by $CH_4$-poor inflow water may deplete the inventory of $CH_4$ in the water column of the EGB. However, due to turbulent mixing at the contact between the inflow and older water masses (Schmale et al., 2016), oxidation of $CH_4$ may also be expected to accelerate the depletion of $CH_4$. In 2015, we observed the gradual erosion of a mid-water $[CH_4]$ maximum between March and August, followed by a re-accumulation of $CH_4$ in both mid- and deep-water towards the end of the year (Fig. 2). To estimate the relative effects of displacement and oxidation on the $CH_4$ inventory during the studied period, we compared the $CH_4$ inventory with that of phosphate ($PO_4^{3-}$) at BY15 (Fig. 3). Changes in deep water $[PO_4^{3-}]$ in the EGB following an MBI have been shown to be predominantly controlled by displacement (Schneider, 2011), due to the fact that a stagnant $PO_4^{3-}$-rich bottom water mass is displaced by a $PO_4^{3-}$-poor inflow, and comparatively little $PO_4^{3-}$ is sequestered into sediments over the following months despite the expansion of oxic





conditions. The logic of the apparent slow response in the redox-sensitivity of $PO_4^{3-}$ is that $PO_4^{3-}$ sequestration requires the presence of solid-phase iron oxyhydroxides, which are not available in the open water column where the majority of the $PO_4^{3-}$ is located.

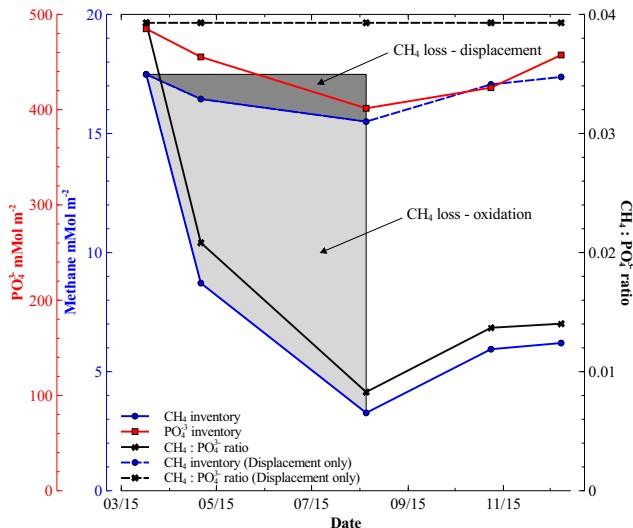

**Figure 3.** Evolution of the inventories of $CH_4$ (solid blue line with dots) and $PO_4^{3-}$ (solid red line with squares) below 70 m at BY15, and the molar inventory ratio $CH_4:PO_4^{3-}$ over the year 2015 (solid black line with crosses). Results from all measured depths at all sampling times were included in the integration, with the exception data from June, which was omitted due to the loss of deep water $CH_4$ samples (Table S1). Pictured also is the hypothetical evolution of the $CH_4$ inventory (dashed blue line with dots) if the $CH_4:PO_4^{3-}$ ratio (dashed black line with crosses) had remained constant during the year, i.e. the effect of water mass displacement was equal for $CH_4$ and $PO_4^{3-}$. The lighter grey area represents the estimated loss of $CH_4$ due to oxidation and the darker grey represents the loss due to displacement. Note the different scales of the y-axes.

Schneider (2011) estimated that two-thirds of the decline in deep water $[PO_4^{3-}]$ in the EGB following the MBI of 2003 could be attributed to displacement. If displacement had accounted for a similar proportion of the decline in both the $CH_4$ and $PO_4^{3-}$ inventories during March to August 2015, it follows that the molar ratio of $PO_4^{3-}$ to $CH_4$ would have remained constant during this period. Instead, we observe that the inventory ratio of $PO_4^{3-}:CH_4$ declined rapidly in the first half of the year, implying a far stronger impact of oxidation on $CH_4$ than on $PO_4^{3-}$. Indeed, the $CH_4$ inventory was depleted from ~18 to ~3 mMol $m^{-2}$ over this interval, while the $PO_4^{3-}$ inventory declined only slightly, from ~480 to ~400 mMol m-2 (Fig. 3).These results provide strong evidence for significant aerobic oxidation of $CH_4$ in the water column as a consequence of the MBI, which concurs with a recent study in which *in situ* oxidation rates were measured (Schmale et al., 2016).

When expressed as volume weighted averages (below 70 m), the observed decline in $[CH_4]$ over March to August 2015 was from 108 nM in March to 16.3 nM in August. This is a period of 141 days, giving a total rate of loss of 0.65 nM $d^{-1}$, of which 0.51 nM $d^{-1}$ (79 %) is potentially due to oxidation, based on the results shown in Fig. 3. Schmale et al. (2016) reported





CH$_4$ oxidation rates of up to 0.9 nM d$^{-1}$ and elevated cell counts for methane-oxidizing bacteria at central EGB in March 2015. Such oxidation rates are 2–10 times higher compared to stagnation conditions (Jakobs et al., 2014). The introduction of a second, deeper redoxcline and active turbulent mixing processes clearly accelerate CH$_4$ oxidation following an MBI by enhancing the volume of water in which CH$_4$ and O$_2$ come into contact. It should be noted, however, that the Schmale et al. (2016) study focused specifically on high turbulence transition zones and thus reports localized maximum estimates of CH$_4$ oxidation rates. In contrast, our study provides a first order estimate of the bulk oxidation rate the sub-halocline water column at site BY15.

### 4.3 N$_2$O dynamics in the EGB

Under stagnation conditions, the deep anoxic waters of the EGB are almost entirely devoid of N$_2$O (Brettar and Rheinheimer, 1991), but the hypoxic margins are hotspots for N$_2$O production, similar to oceanic oxygen minimum zones (Babbin et al., 2015). The accumulation of N$_2$O over time after an MBI is related to the formation of large hypoxic water masses. N$_2$O production is highest under nearly anoxic conditions, where oxygen concentrations restrain both nitrification (too little oxygen) and denitrification (too much oxygen). Both processes produce N$_2$O, with a higher N$_2$O yield in oxygen stress (Goreau et al., 1980; Patureau et al., 1994; Ji et al., 2015). In addition, when nitrifying microbes become oxygen limited, they too, start to reduce NO$_2^-$ to N$_2$O (Poth and Focht, 1985; Wilson et al., 2014).

For the most part, the N$_2$O concentrations measured in the oxic waters of this study were between 10–20 nM, which is well in agreement with previous studies (Bange, 2006; Walter et al., 2006). The volume-weighted average [N$_2$O] below 70 m depth at BY15 increased from 11.8 nM to 24.4 nM from March to August (141 d), giving a net increase rate of 0.09 nM d$^{-1}$. Walter et al. (2006) reported a similar increase rate of 0.105 nM d$^{-1}$ from the whole water column below 70 m in the EGB, over a period of 167 days after the 2003 MBI, which they ascribed largely to nitrification. We observed a decline in [NH$_4^+$] in the mid-water layer at BY15 over the period March to August 2015 (Fig. 4). The decline in [NH$_4^+$] resembled the loss of CH$_4$ over the same period, and was coupled to increasing [NO$_2^-$] and [NO$_3^-$] (Fig. 4). These observations strongly suggest that oxidation of NH$_4^+$ (i.e. nitrification) following the MBI was the main pathway of N$_2$O production during the first half of 2015.

One of the most interesting observations in this study was the extremely high [N$_2$O] (877 nM) measured at 225 m at BY15 in October, which is to our knowledge the second highest value ever reported from the Baltic Sea (Rönner (1983) reported 1523 nM at BY38 bottom water in the WGB), and several times higher than the concentrations typically found from the Baltic (Brettar and Rheinheimer, 1991; Bange et al., 1996; Bange, 2006; Walter et al., 2006). The oxygen concentration at the depth of the extreme [N$_2$O] in this study was below 1 µM, which is comparable to values previously observed in settings of high N$_2$O production elsewhere in the ocean (Naqvi et al., 2010; Babbin et al., 2015). The large drop in the [NO$_2^-$] and [NO$_3^-$] below 200 m at BY15 from October onwards (Fig. 4) suggests that the rate of benthic denitrification increased towards the end of 2015. Simultaneously increasing [NH$_4^+$] indicate a slowing down of rates of nitrification and possibly enhanced DNRA (Jäntti and Hietanen, 2012). Hence, the transitional conditions between nitrification and denitrification regimes towards the end of 2015 appear to have favored an extreme, short-lived accumulation of N$_2$O in the deep waters of the EGB. This may be seen as a delayed, but important consequence of the MBI on nitrogen cycling in the Baltic.



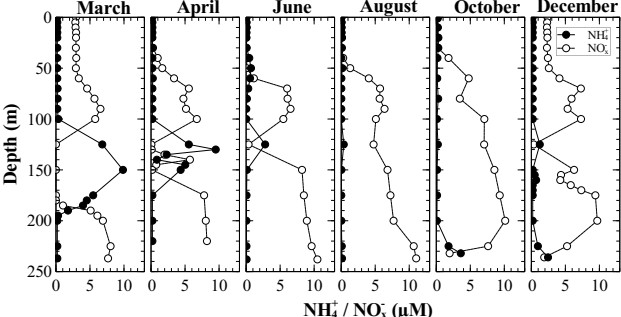

**Figure 4.** Ammonium ($NH_4^+$, filled circles, solid line) and combined nitrite-nitrate ($NO_x^-$, open circles, dotted line) concentrations (μM) at BY15 over the whole sampling period.

### 4.4 Processes at the northern limit of the MBI

As MBIs progress northwards, the density differences between the old and new water masses are weakened and the interactions become more complex and difficult to predict (Eilola et al., 2014). Inflowing water masses detach into intrusive layers (Baines, 2001) which may interact chaotically under turbulent flow. Site BY20 is situated near the northern margin of the EGB and represents the northernmost limit of the observable effects of the 2014 MBI. In this zone, various physical factors, e.g. turbulent mixing and shearing, internal waves, and boundary waves breaking against the sloping seabed (Reissmann et al., 2009; Eilola et al., 2014; Schmale et al., 2016) likely created a complex and temporally variable vertical zonation of redox conditions during 2015. Despite the persistence of $H_2S$ at BY20 throughout the year, the $CH_4$ and $N_2O$ distributions are highly variable (Fig. 2), suggesting an impact of oxidation and reduction processes related to the inflow. For example, a large enrichment of $N_2O$ was observed in the near bottom water of BY20 in October, similar to that observed at BY15, and the $CH_4$ and $N_2O$ distributions generally anti-correlate as observed throughout our data from all stations. We interpret the observed patterns at BY20 as evidence for a dynamic water column at this site, with mobile interleaved layers which may carry a displaced signal of redox processes occurring further south (e.g. $CH_4$ oxidation, nitrification).

### 5 Conclusions

The Major Baltic Inflow of 2014 caused considerable changes in oxygen conditions of the Eastern Gotland Basin, which had extensive effects on the $CH_4$ and $N_2O$ dynamics of the Basin. $CH_4$ mostly disappeared from the eastern basin during the first half of 2015, mainly due to oxidation following turbulent mixing between old and new water masses. However, $CH_4$ began to accumulate again by the end of the year, as deep water conditions reverted to anoxia. Enhanced $N_2O$ production was evident throughout the Eastern Gotland Basin during 2015, attributed primarily to nitrification as a consequence of the MBI. Extreme values of $N_2O$ near the seafloor in late 2015 were likely caused by a combination of nitrification and denitrification under transitional conditions. The northern limit of the effect of the MBI on $CH_4$ and $N_2O$ dynamics appears to have been the

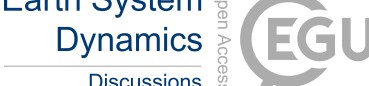

northern part of the Eastern Gotland Basin, and even here direct oxygenation of the deep waters was not observed. The Western Gotland Basin $CH_4$ and $N_2O$ biogeochemistry was not influenced by the inflow at any point during the study period.

*Acknowledgements.* This research was funded by the Academy of Finland projects 139267, 272964 and 267112. The research leading to these results has also received funding from the European Union Seventh Framework Programme (FP7/2007-2013) under grant agreement n° 312762. We would like to extend our thanks to SYKE, SMHI and Tallinn University of Technology for the supplementary data and for allowing us on board their cruises, and to the crews and captains of R/V Aranda and R/V Salme.



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
