# Peer review of "Figure S1. Establishment of cut-off concentrations of $\text{CH}_4$ (left) and $\text{N}_2\text{O}$ (right), below which measured sample values are regarded as indistinguishable from those of blank pre-evacuated Exetainers. Yellow points represent measured peak areas of $\text{CH}_4$ and"

_Earth System Dynamics, 2017_

## Referee Comment (RC1) · Anonymous Referee #1 · 22 May 2017

The manuscript "Effects of the 2014 Major Baltic Inflow on methane and nitrous oxide dynamics in the water column of the Central Baltic Sea" by Myllykangas et al. presents measurements of N2O and CH4 in the deep waters of the central Baltic Sea during an inflow event of saline, oxygen-rich waters.

The manuscript presents valuable insights into the dynamics of N2O and CH4 accumulation and depletion in the water column as a consequence of water column oxygenation and subsequent reestablishment of anoxic conditions. The high temporal resolution of the samplings allows a comprehensive reconstruction of the N2O and CH4 development and the processes influencing their distribution.

The manuscript is generally well structured and written and I can support its publication

with only minor corrections:

P. 2, lines 15-17: I would emphasize the role of oxygen on N2O production and consumption processes.Given the fact that oxygen concentrations show strong gradients and a large variability during the Major Baltic Inflow event, the sensitivity of N2O production of nitrification and denitrification should be discussed more detailed.

P. 3, lines 16-17 & P. 4, lines 1-4: please explain in more detail how the samples were drawn from the Niskin bottles and transferred into the Exetainers. Were the syringes directly connected to the Niskin bottles? How was the system flushed to enable bubble-free sampling of the Niskin bottles? How exactly were the samples transferred to the Exetainers and how was air contamination prevented?

P. 4, line 23: Equation (1) is not entirely correct: Patm in the first term of the equation has to be given in Pascal since this term represents the ideal gas law.

P. 9, lines 14-15: please add Löscher et al. (2012) as a reference for the oxygen-dependency of N2O production by nitrifying archaea: Löscher, C. R., Kock, A., Könneke, M., Laroche, J., Bange, H. W. and Schmitz, R. A.: Production of oceanic nitrous oxide by ammonia-oxidizing archaea, Biogeosciences, 9(7), 2419–2429, 2012.

---

## Referee Comment (RC2) · Anonymous Referee #2 · 29 May 2017

Review: "Effects of the 2014 Major Inflow on methane and nitrous oxide dynamics in the water column of the Central Baltic Sea" by J.-P. Myllykangas et al., 2017

Summary: The paper describes the MBI event of 2014 and its impact on the concentration patterns of N2O and CH4 in the Baltic Sea, measured at 5 stations in the Gotland Basin. The findings and conclusions are consistent with previous results. This study contributes to the monitoring of greenhouse gas behavior under changing conditions and is therefore worth to be published. Although monitoring is not delivering the most exciting science on a short-term, it is indispensable for basic understanding in a long-term point of view and prediction of future changes. I recommend publishing with some minor corrections.

[Figure]

Comments: The manuscript is in general well written and structured. Some parts could be more detailed. The literature cited is mostly adequate, citations need to be checked as discussion papers might be already published (see Bange et al. 2009, Walter et al. 2006).

The abstract is quite general; for a first overview it might be helpful to also include main results and numbers.

The Introduction part describes the processes very briefly and focus on the main pathways. For several statements (e.g. page 2, line 3-30) an additional citation of more recent publications would be preferable, as well as a more detailed description. The sentence "Both advective processes . . .. (page 2 / 3, line 30 / 1) belongs more to the conclusions than to the introduction.

In the Material and Methods part please include a detailed error description and estimation, especially with view on the gas transfers between several plastic syringes. Please explain the advantages of storing the samples as described. Check the formula and its units. Please include information why those 5 stations have been chosen and link them to previous Baltic Sea monitoring programmes.

In the Discussion part the CH4 dynamics in the EGB are not very clear described. Especially the CH4:PO4 ratio approach could be more detailed as the figure is relatively complex.

Most of the figures are too small and included information is hardly readable. The information in the figure captures might be shortened or included in the text. Figure 4: if NH4+ and NOx- have been measured at more than one station (BY15), it would be helpful to include the information into the overview Figure 2.

The information given in the supplementary part could be better introduced and referred to in the manuscript.
* * *

---

## Author Comment (AC1) · 22 Jun 2017

P. 2, lines 15-17: I would emphasize the role of oxygen on N2O production and consumption processes. Given the fact that oxygen concentrations show strong gradients and a large variability during the Major Baltic Inflow event, the sensitivity of N2O production of nitrification and denitrification should be discussed more detailed.

> *We will revise the text accordingly and add more detail about the role of oxygen in N2O production.*

P. 3, lines 16-17 & P. 4, lines 1-4: please explain in more detail how the samples were drawn from the Niskin bottles and transferred into the Exetainers. Were the syringes directly connected to the Niskin bottles? How was the system flushed to enable bubblefree sampling of the Niskin bottles? How exactly were the samples transferred to the Exetainers and how was air contamination prevented?

> *Syringes were connected directly to the Niskin bottle bottom valve with a short rubber tube. Prior to connecting the syringe, the tube was flushed several times its volume with sample water and care was taken to squeeze out any visible bubbles from the tube.*

*After equilibration, the original sampling syringe was connected to a dry syringe via a 3-way stopcock and from the dry syringe gas injected to an exetainer with a short needle. The syringes were always kept closed with a stopcock to minimise air contamination.*

*More detail about the sampling process will be included in the revised version of the manuscript.*

P. 4, line 23: Equation (1) is not entirely correct: Patm in the first term of the equation has to be given in Pascal since this term represents the ideal gas law.

> *All units of the equation will be converted to SI units in the revised version of the manuscript.*

P. 9, lines 14-15: please add Löscher et al. (2012) as a reference for the oxygen dependency of N2O production by nitrifying archaea: Löscher, C. R., Kock, A., Könneke, M., Laroche, J., Bange, H. W. and Schmitz, R. A.: Production of oceanic nitrous oxide by ammonia-oxidizing archaea, Biogeosciences, 9(7), 2419–2429, 2012.

> *Text will be revised accordingly and the reference will be added.*

---

## Author Comment (AC2) · 22 Jun 2017

Comments: The manuscript is in general well written and structured. Some parts could be more detailed. The literature cited is mostly adequate, citations need to be checked as discussion papers might be already published (see Bange et al. 2009, Walter et al. 2006).

> *The referee is correct, both of the above studies have already been published and reference will be updated accordingly.*

The abstract is quite general; for a first overview it might be helpful to also include main results and numbers.

> *More detail will be added to the abstract, along the main results and numbers.*

The Introduction part describes the processes very briefly and focus on the main pathways. For several statements (e.g. page 2, line 3-30) an additional citation of more recent publications would be preferable, as well as a more detailed description.

> *The text will be revised and the main processes will be described in greater detail with more up to date citations where possible. The concise style of the introduction was a conscious decision in keeping with relative short format of the article.*

The sentence "Both advective processes.." (page 2 / 3, line 30 / 1) belongs more to the conclusions than to the introduction.

> *Our aim was to investigate the various factors potentially contributing to removal of methane and feel that it is therefore relevant to introduce the main mechanisms in the beginning.*

In the Material and Methods part please include a detailed error description and estimation, especially with view on the gas transfers between several plastic syringes.

> *A more detailed error description will be included in the revised manuscript, including potential error caused by the syringe transfer.*

Please explain the advantages of storing the samples as described.

> *The main advantage of transferring the samples into Exetainers already onboard is that they samples can be subsequently transported with ease to be analysed in the home lab. Sturm et al. (2015, Limnology and Oceanography: Methods, Vol 13, Issue 7) have shown that there is no noticeable loss of gas from exetainers over several weeks of storage.*

Check the formula and its units.

> *The formula has been checked and the manuscript will be updated to use SI units.*

Please include information why those 5 stations have been chosen and link them to previous Baltic Sea monitoring programmes.

> *All five stations have been consistently monitored for several decades for basic hydrographic parameters and have been used in numerous previous studies. The fieldwork for this study was conducted partly during monitoring cruises, hence the choice of stations was somewhat fixed. However, these stations represent well the conditions in both the eastern and western basin well and had the best monitoring coverage out of all stations based in the Gotland basin during the year after the inflow.*

In the Discussion part the CH4 dynamics in the EGB are not very clear described. Especially the CH4:PO4 ratio approach could be more detailed as the figure is relatively complex.

> *We appreciate that this section is rather complex and we took care to describe our approach as clearly as possible. However, we take on board the reviewer's comment and try to improve the description of our approach and our interpretations of the results.*

Most of the figures are too small and included information is hardly readable. The information in the figure captures might be shortened or included in the text.

> *Font sizes of all figures will be increased and captions will be shortened where possible.*

Figure 4: if NH4+ and NOx- have been measured at more than one station (BY15), it would be helpful to include the information into the overview Figure 2.

> *Both data are available. However, these data will be presented with greater detail in an upcoming companion paper and therefore we chose not to present them here.*

The information given in the supplementary part could be better introduced and referred to in the manuscript.

> *The supplementary material will be re-structured and revised.*

---

## Author Response (AR1)

**Reviewer I comments:**

P. 2, lines 15-17: I would emphasize the role of oxygen on N2O production and consumption processes. Given the fact that oxygen concentrations show strong gradients and a large variability during the Major Baltic Inflow event, the sensitivity of N2O production of nitrification and denitrification should be discussed more detailed.

- *P. 2, line 15-30: the description of N2O production has been extended and new references have been added*

P. 3, lines 16-17 & P. 4, lines 1-4: please explain in more detail how the samples were drawn from the Niskin bottles and transferred into the Exetainers. Were the syringes directly connected to the Niskin bottles? How was the system flushed to enable bubblefree sampling of the Niskin bottles? How exactly were the samples transferred to the Exetainers and how was air contamination prevented?

- *P. 3, line 32: added "... with a short rubber tube  …"*
- *P. 4, line 2: added "through a stopcock"*
- *P. 4, line 4: added  "Care was taken …"*

P. 4, line 23: Equation (1) is not entirely correct: Patm in the first term of the equation has to be given in Pascal since this term represents the ideal gas law.

- *P. 5, line 1: Clarified the calculation and added the missing step that explains the origin of Patm.*

P. 9, lines 14-15: please add Löscher et al. (2012) as a reference for the oxygen dependency of N2O production by nitrifying archaea: Löscher, C. R., Kock, A., Könneke, M., Laroche, J., Bange, H. W. and Schmitz, R. A.: Production of oceanic nitrous oxide by ammonia-oxidizing archaea, Biogeosciences, 9(7), 2419–2429, 2012.

- *P. 10, line 13: Reference added, also to the introduction*

**Reviewer II comments:**

Comments: The manuscript is in general well written and structured. Some parts could be more detailed. The literature cited is mostly adequate, citations need to be checked as discussion papers might be already published (see Bange et al. 2009, Walter et al. 2006).

- *Both references have been updated*

The abstract is quite general; for a first overview it might be helpful to also include main results and numbers.

- *The abstract has been slightly expanded and the main results have been included*

The Introduction part describes the processes very briefly and focus on the main pathways. For several statements (e.g. page 2, line 3-30) an additional citation of more recent publications would be preferable, as well as a more detailed description.

- *While the concise style of the introduction was a conscious decision in keeping with relative short format of the article, the description of N2O production has been extended and new references have been added, also as requested by reviewer #1.*

The sentence "Both advective processes.." (page 2 / 3, line 30 / 1) belongs more to the conclusions than to the introduction.

- *Our aim was to investigate the various factors potentially contributing to removal of methane and feel that it is therefore relevant to introduce the main mechanisms in the beginning.*

In the Material and Methods part please include a detailed error description and estimation, especially with view on the gas transfers between several plastic syringes.

- *More detail about the sources and amounts of error have been added to the materials and methods section and a more in-depth description about the sources of error has been included in supplement text S1.*

Please explain the advantages of storing the samples as described.

- *The main advantage of transferring the samples into Exetainers already onboard is that they samples can be subsequently transported with ease to be analysed in the home lab. Sturm et al. (2015, Limnology and Oceanography: Methods, Vol 13, Issue 7) have shown that there is no noticeable loss of gas from exetainers over several weeks of storage.*

Check the formula and its units.

- *The calculations have been updated and clarified*

Please include information why those 5 stations have been chosen and link them to previous Baltic Sea monitoring programmes.

- *All five stations have been consistently monitored for several decades for basic hydrographic parameters and have been used in numerous previous studies. The fieldwork for this study was conducted partly during monitoring cruises, hence the choice of stations was somewhat fixed. However, these stations represent well the conditions in both the eastern and western basin well and had the best monitoring coverage out of all stations based in the Gotland basin during the year after the inflow.*
- *P. 3, line 23, added: "All five stations …"*

In the Discussion part the CH4 dynamics in the EGB are not very clear described. Especially the CH4:PO4 ratio approach could be more detailed as the figure is relatively complex.

- *P. 9, line 5 reworded the original text to make it more accessible.*

Most of the figures are too small and included information is hardly readable. The information in the figure captures might be shortened or included in the text.

- *Font sizes of all figures will be increased*
- *Caption for Fig 2. Has been shortenedss*
- *Fig 1. (Map) has been updated*

Figure 4: if NH4+ and NOx- have been measured at more than one station (BY15), it would be helpful to include the information into the overview Figure 2.

- *Both data are available. However, these data will be presented with greater detail in an upcoming companion paper and therefore we chose not to present them here.*

The information given in the supplementary part could be better introduced and referred to in the manuscript.

- *The supplementary material has been reorganised, expanded and references in the text have been revised.*

[revised manuscript text omitted]

---

## Author Response (AR2)

Would you please check definition for the "beta" in formula (1). In Si, Bunsen solvability coefficient should be dimensionless. The one in (1) should have some pressure units in denominator but on page 5 line 2 it is in $10^{**}(-9)$ mol/L.

> *The editor is correct, the original Bunsen solubility coefficient is indeed dimensionless, though sometimes the derived solubility concentrations with units are also called β in the literature. The constants for calculating the solubility given in Wiesenburg & Guinasso (1979) and Weiss & Price (1980) are for 1 atm, so the unis can an also be expressed as mol $L^{-1}$ $atm^{-1}$, though Wiesenburg & Guinasso do not explicitly do so. However, to avoid confusion we've now changed the symbol from β to F as per Weiss & Price (1980), who also give the unit explicitly in mol $L^{-1}$ $atm^{-1}$.*

Also please remove pV=nRT formula on the line 16 page 4 (as you need to define all the ingredients which would be rather redundant).

> *The formula has been removed.*